Advances in drug resistance and resistance mechanisms of four colorectal cancer-associated gut microbiota

Gan Yu 1
Yang Hao 2
Wang Maijian 3
Li Jida 1 4 lijida198485@163.com
1 Institute of Zoonosis, College of Public Health, Zunyi Medical University , Zunyi, Guizhou , China
2 Xingyi City Disease Prevention and Control Center (Municipal Health Supervision Station) , Xingyi, Guizhou , China
3 Institute of Gastrointestinal, Affiliate Hospital of Zunyi Medical University , Zunyi, Guizhou , China
4 Key Laboratory of Maternal & Child Health and Exposure Science of Guizhou Higher Education Institutes , Zunyi, Guizhou , China
Mattei Fabrizio
Electronic publication date: 2025 Jun 10
Publication date: 2025
Volume: 13
Electronic Location ID: e19535
Received 2024 Nov 13; Accepted 2025 May 7
Copyright: © 2025 Gan et al.
Copyright year: 2025
Copyright holder: Gan et al.
License: This is an open access article distributed under the terms of the Creative Commons Attribution License, which permits unrestricted use, distribution, reproduction and adaptation in any medium and for any purpose provided that it is properly attributed. For attribution, the original author(s), title, publication source (PeerJ) and either DOI or URL of the article must be cited.
License URL: https://creativecommons.org/licenses/by/4.0/

Keywords: Colorectal cancer, Bacteroides fragilis, Fusobacterium nucleatum, Enterotoxin-producing Escherichia coli, Enterococcus faecalis, Drug resistance mechanism

Funding: Natural Science Research Program of Guizhou Provincial Department of Education 2024 QJJ [2024] 335 Zunyi Science and Technology Program Project Zunshi Kehe HZ Zi [2024] No. 303 Special Project for Cultivation of New Academic Talents and Innovation Exploration of Zunyi Medical University [2021] 1350-038 Science and Technology Fund Project of Guizhou Health Care Commission gzwjkj2019-1-123 Governor’s Special Fund for Outstanding Scientific and Technological Education Talents in Guizhou Province [2011]57 Scientific Research Program of Guizhou Provincial Department of Education QJJ [2023] 019 This work was supported by Natural Science Research Program of Guizhou Provincial Department of Education 2024(QJJ [2024] 335), Zunyi Science and Technology Program Project (Zunshi Kehe HZ Zi [2024] No. 303), Special Project for Cultivation of New Academic Talents and Innovation Exploration of Zunyi Medical University for the year 2021 (Qianke Platform Talents [2021] 1350-038), Science and Technology Fund Project of Guizhou Health Care Commission (No. gzwjkj2019-1-123), Governor’s Special Fund for Outstanding Scientific and Technological Education Talents in Guizhou Province (No. [2011]57), Scientific Research Program of Guizhou Provincial Department of Education (QJJ [2023] 019). The funders had no role in study design, data collection and analysis, decision to publish, or preparation of the manuscript.

==============================
Colorectal cancer (CRC) is a common malignant tumor in the gastrointestinal tract with inconspicuous early symptoms, high morbidity and mortality, and poor prognosis. Gut microbiota are present in the human intestinal system and have certain functions, which include the integrity of the epithelial barrier and the enhancement of protective immune responses. The etiology of CRC is numerous and complex, including poor lifestyle and dietary habits, and instability of the gut microbiota, which is considered to be one of the major factors in the development of CRC, includes mainly Bacteroides fragilis, Fusobacterium nucleatum, Escherichia coli, and Enterococcus faecalis. Enrichment of these bacteria in CRC tumor tissues may increase other pro-inflammatory opportunistic pathogens and decrease butyrate-producing bacteria, leading to an imbalance in intestinal homeostasis (dysbiosis) and ultimately tumor formation. Antibiotic-induced changes in the gut microbiota affect tissue utilization and redox homeostasis of macronutrients and micronutrients. However, the long-term use and abuse of antibiotics has made the problem of drug resistance a difficult problem that currently plagues the regulation of gut microbiota, as well as a complicated issue in the prevention and treatment of CRC. In this review, we elucidated the drug resistance of four CRC-associated gut microbiota, namely Bacteroides fragilis, Fusobacterium nucleatum, Escherichia coli, and Enterococcus faecalis, and discussed the common and different aspects of the resistance mechanisms of the four gut microbiota, with the aim of providing a basis for the prevention and control of CRC.

Introduction

In recent years, the incidence and mortality of colorectal cancer (CRC) have been high (Fig. 1 (Bray et al., 2018; Fitzmaurice et al., 2019; Kocarnik et al., 2022; Sung et al., 2021)). In April 2024, the International Agency for Research on Cancer (IARC) published the latest global cancer statistics for 2022 in the CA: A Cancer Journal for Clinicians, which showed that there were approximately 1,926,000 CRC cases and 904,000 deaths in 2022 (Hossain et al., 2022; Yuxin, Kaifeng & Wenqing, 2024). The pathogenesis and progression of CRC involves a number of mechanisms, such as abnormal cell proliferation and differentiation, invasion of neighboring tissues and distant metastasis, and a series of pathophysiological mechanisms, in which many genes and signaling pathways are involved in the pathogenesis and progression of CRC, but the etiology of CRC is unclear (Ionescu et al., 2023), this is probably the main reason for the high prevalence of this type of disease, as the unknown etiology limits the effectiveness of its prevention and treatment strategies, leading to high morbidity and mortality rates.

Figure 1 Changes in global colorectal cancer incidence and deaths, 2017–2022.

As early as the 1970s, studies have found that gut microbiota are closely related to the development of CRC (Mentella et al., 2020). In recent years, the correlation between gut microbiota profiles and colorectal adenoma-cancer sequences has been validated based on high-throughput sequencing and population-based big data analysis, further confirming that changes in gut microbiota play an important role in CRC development and progression (Tilg et al., 2018). Among them, Bacteroides fragilis, Fusobacterium nucleatum, Escherichia coli, and Enterococcus faecalis are most closely associated with the development of CRC (Bonnet et al., 2014; Feng et al., 2015; Nakatsu et al., 2015; Williamson et al., 2022), The ability to effectively regulate the abundance of Bacteroides fragilis, Fusobacterium nucleatum, Escherichia coli, and Enterococcus faecalis in the intestinal tract has become a key component in the prevention and control of CRC. Along with the long-term use and abuse of antibiotics, the problem of drug resistance has become a difficult problem in the regulation of gut microbiota, and also a complicated issue in the prevention and control of CRC. Antibiotic use can increase the abundance of antibiotic-resistant bacterial species and the abundance of antibiotic-resistant genes in the gut microbiota. There is growing evidence that prolonged, frequent and/or combined use of antibiotics may also be a risk factor for CRC (Sanyaolu et al., 2020). It was found that there are more than 20 bacterial species in the gut microbiota enriched in the CRC, of which more than 10 carry antibiotic resistance genes, including Escherichia coli, Bacteroides fragilis, and Fusobacterium nucleatum, with Escherichia coli acting as an important carrier of resistance genes (Liu et al., 2021). It has been shown that high levels of antibiotic exposure are positively correlated with CRC progression, and that those with the highest antibiotic exposure are at the greatest risk of developing CRC compared to those who use less antibiotics, and we hypothesized whether this situation is due to the severe antibiotic resistance that develops in gut microbiota that receive large amounts of antibiotics, and therefore, the use of antibiotics affects changes in gut microbiota in response to changes in the process of CRC development (Aneke-Nash et al., 2021). The aim of this review is to provide an in-depth discussion on the drug resistance of four colorectal cancer-associated gut microbiota, namely Bacteroides fragilis, Fusobacterium nucleatum, Escherichia coli, and Enterococcus faecalis, as well as the mechanisms of drug resistance, with the aim of providing a rationale for the prevention and control of CRC.

Survey methodology

This review is the result of a systematic literature search on PubMed and Web of Science. It aimed to find articles related to drug resistance as well as mechanisms of resistance in four CRC-associated gut microbiota, namely Bacteroides fragilis, Fusobacterium nucleatum, Escherichia coli, and Enterococcus faecalis, for the period up to 2024. Articles used different combinations of search terms including “colorectal cancer or colorectal tumor or CRC”, “(gastrointestinal microbiome or gut microbiota) and (colorectal cancer or colorectal tumor or CRC)”, “(Bacteroides fragilis or BF) and (colorectal cancer or colorectal tumor or CRC)”, “Escherichia coli and (colorectal cancer or colorectal tumor or CRC)”, “(Fusobacterium nucleatum or Fn) and (colorectal cancer or colorectal tumor or CRC)”, “Enterococcus faecalis and (colorectal cancer orcolorectal tumor or CRC)”, “(Bacteroides fragilis or BF) and antibiotics”, “Escherichia coli and antibiotics”, “(Fusobacterium nucleatum or Fn) and antibiotics”, “Enterococcus faecalis and antibiotics”, “(Bacteroides fragilis or BF) and drug resistance”, “Escherichia coli and drug resistance”, “(Fusobacterium nucleatum or Fn) and drug resistance”, “Escherichia coli and drug resistance”, “Enterococcus faecalis and drug resistance”, “(Bacteroides fragilis or BF) and drug resistance mechanisms”, “Escherichia coli and resistance mechanisms”, “(Fusobacterium nucleatum or Fn) and resistance mechanisms”, and “Enterococcus faecalis and resistance mechanisms”. Meanwhile, we reviewed the main resistance mechanisms of the four colorectal cancer-associated gut microbiota, namely Bacteroides fragilis, Fusobacterium nucleatum, Escherichia coli, and Enterococcus faecalis, “(Bacteroides fragilis or BF) and drug resistance genes”, “(Bacteroides fragilis or BF) and horizontal gene transfer”, “(Bacteroides fragilis or BF) and efflux pumps”, “(Fusobacterium nucleatum or Fn) and drug resistance genes”, “(Fusobacterium nucleatum or Fn) and horizontal gene transfer”, “(Fusobacterium nucleatum or Fn) and efflux pumps”, “Escherichia coli and drug resistance genes”, “Escherichia coli and horizontal gene transfer”, “Escherichia coli and efflux pumps”, “Enterococcus faecalis and drug resistance genes”, “Enterococcus faecalis and horizontal gene transfer”, “Enterococcus faecalis and the efflux pump”, including Bacteroides fragilis, Fusobacterium nucleatum, Escherichia coli, and Enterococcus faecalis with β-lactamase, nim, cfiA, cepA, cfxA, tetQ, ermF, and other resistance genes. We used a search strategy to obtain the titles and abstracts of the relevant studies that we initially screened and retrieved the full text. We also reviewed relevant references in the articles to ensure the comprehensiveness of the studies.

Overview of colorectal cancer-associated enteric bacteria

In recent years, Fusobacterium nucleatum (F. nucleatum), pathogenic Escherichia coli (E. coli), enterotoxigenic Bacteroides fragilis (B. fragilis), and Enterococcus faecalis (E. faecalis) have been most widely reported in CRC-associated gut microbiota (Fig. 2 (Dougherty & Jobin, 2023; Song, Chan & Sun, 2020)). Epidemiologic studies have found a significant positive correlation between F. nucleatum abundance and CRC stage, preferentially enriched in advanced CRC tissues (Dougherty & Jobin, 2023; Xu et al., 2021a). However, as the gut microbiota is one of the early biomarkers of cancer, our study also provides recommendations for the occurrence and development of cancer. According to research reports, the positivity of F. nucleatum in precancerous lesions is significantly correlated with high CpG island methylation phenotype (CIMP) and larger tumor size, and there are also research reports that F. nucleatum is considered a potential biomarker for predicting the prognosis of patients with proximal CRC (Ito et al., 2015; Quaglio et al., 2022). The main mechanisms by which F. nucleatum promotes CRC progression are related to its adhesin molecule FadA, and dysregulation of the Wnt signaling pathway, in which β-catenin forms a protein complex with FadA, Annexin A1 protein, and E-cadherin protein, is associated with CRC progression changes (Rubinstein et al., 2019). The current study demonstrates that E. coli Nissle 1917, a natural intestinal probiotic, has anti-inflammatory properties and can reduce colorectal adenoma load and alter the tumor immune microenvironment (Gurbatri et al., 2024). However, pathogenic Escherichia coli was more prevalent in stool or tissue samples from patients with CRC compared to patients with inflammatory bowel disease (IBD) or healthy controls (Dubinsky, Dotan & Gophna, 2020). E. coli produces colibactin, encoded by the pks gene, which has a variable and complex pathway in promoting the progression of CRC, including genotoxicity by causing double-stranded breaks in cellular DNA; it can also lead to disruption of the normal intestinal barrier as well as inflammation; and it can interfere with the normal cell cycle, leading to the death of normal cells or the transformation of abnormal cells, which can lead to the onset of CRC (Sadeghi, Mestivier & Sobhani, 2024). In most studies that used tissue biopsies and assessed the prevalence of enterotoxigenic B. fragilis (ETBF) by quantitative real-time PCR, ETBF abundance was either positively correlated with tumor stage or equally enriched throughout the tumor stage (Dougherty & Jobin, 2023; Song, Chan & Sun, 2020). ETBF produces a B. fragilis toxin (BFT), which can specifically bind to intestinal epithelial receptors, thereby activating the Wnt and NF-κB signaling pathways, increasing cell proliferation, epithelial release of pro-inflammatory mediators, and inducing DNA damage, thereby affecting the development of CRC (Boleij et al., 2015). E. faecalis is one of the most common commensal enterococci found in human feces, and studies have shown that it induces migration and invasion of CRC cells (Williamson et al., 2022). The oncogenic pathway of E. faecalis is mainly related to the formation of reactive oxygen species (ROS), and it has been shown that the enrichment of E. faecalis leads to an increase in the expression of COX-2 in macrophages, which in turn induces the production of ROS, thus leading to chromosomal instability and ultimately promoting the progression of CRC (Tortora et al., 2022).

Figure 2 Literature reports of gut microbiota associated with colorectal cancer in the last decade.

Dysregulation of the gut microbiota due to the development of CRC, regulating the homeostasis of the gut microbiota is a promising therapy, and existing treatments focus on antagonizing the growth of pathogenic bacteria through probiotics and their metabolites in the intestinal tract, thereby attenuating and delaying the development of CRC (Gou et al., 2023). Studies have shown that CRC patients require adequate restoration of intestinal homeostasis during the post-surgical recovery period and subsequent chemotherapy, and that the administration of probiotics during post-surgical chemotherapy can effectively alleviate gastrointestinal complications and improve microbiota disorders in the gastrointestinal tract of patients (Huang et al., 2023).

Overview of drug resistance in four species of enteric bacteria

Antibiotics were first discovered in the 1920s. The 1950s and 1960s were the golden age of antibiotic development, and to this day, antibiotics are the drug of choice for the clinical treatment of harmful bacteria (Ramirez et al., 2020). Among them, research on antibiotics for the treatment of CRC-associated gut microbiota has never stopped, such as cephalosporin antibiotics, for which five generations of drugs with different antimicrobial activities have been developed (National Institute of Diabetes and Digestive and Kidney Diseases, 2012b). However, with the emergence of antibiotic abuse and other phenomena in recent years, the reports of CRC-associated gut microbiota resistance to antimicrobial drugs have gradually increased. Among the existing reports, the studies on the resistance mechanisms of the above four bacteria are mostly related to their drug-resistant genes. According to research, one of the main reasons for the reduced effectiveness of CRC treatments is the development of antibiotic resistance, which occurs in almost all CRC patients during treatment, and which makes anti-cancer drugs less effective (Hu et al., 2016). Based on this, we first summarized the resistance profiles of the four CRC-associated bacteria mentioned above and the resistance mechanisms mediated by resistance genes, and the results are shown in Table 1 (Alauzet, Lozniewski & Marchandin, 2019; Barlaam et al., 2019; Bartha et al., 2011; Bush & Jacoby, 2010; Conceição et al., 2014; Edwards & Read, 2000; Eitel et al., 2013; Goldstein, 2014; Grossman, 2016; He et al., 2016; Hiraga et al., 2008; Hussain et al., 2021; Huys et al., 2004; Johnsen et al., 2017; Kierzkowska et al., 2019; Li, Walker & De Oliveira, 2022; Naselli et al., 2022; Patel et al., 2023; Schwarz et al., 2021; Tamma et al., 2019; Voha et al., 2006; Wang et al., 2000; Yekani et al., 2022).

Table 1 Resistance profiles and resistance mechanisms of four colorectal cancer-associated enteric bacteria.

Bacteria	Drug class	Major drug resistance genes	Resistance mechanisms	References	
Enterotoxigenic Bacteroides fragilis	Carbapenems	cfiA	The upstream region of the cfiA gene undergoes mutations through insertion sequences (ISs) elements, as ISs induce transcription of cfiA.	Alauzet, Lozniewski & Marchandin (2019)	
	Metronidazole	nim	nimA-positive strains mainly reduce 5-Ni to its amine derivatives, thus avoiding the formation of nitroso groups.	Edwards & Read (2000), Yekani et al. (2022)	
	Tetracycline	tetQ, tetX	The tetX gene encodes an oxidoreductase that inactivates tetracycline under aerobic conditions.	Bartha et al. (2011)	
	Clindamycin	ermB, ermF, mefA	erm genetically determined macrolide lincosamide streptavidin-type methylase, which alters the ribosome and prevents clindamycin from binding efficiently to the ribosome.	Johnsen et al. (2017), Kierzkowska et al. (2019)	
	Lincomycin	linA	linA is an O-nucleotide transferase located in NBU2, which is mainly mobilized into the B. fragilis receptor by proteins encoded on coupled transposons, leading to drug resistance.	Eitel et al. (2013), Wang et al. (2000)	
Fusobacterium nucleatum	β-lactams	blaFUS-1	blaFUS-1 hydrolyzes substrates through the formation of acylases from the active-site serine.	Bush & Jacoby (2010), Voha et al. (2006)	
pathogenic Escherichia coli (E.coli)	β-lactams	CTX-M	Can encode ultra-broad-spectrum β-lactamases, which show resistance to β-lactam antibiotics through hydrolysis.	Hussain et al. (2021), Naselli et al. (2022)	
	Rifampicin	rpoB	Rifampicin binds to the beta subunit of RNA polymerase to inhibit transcription, while substitution in rpoB inhibits rifampicin binding.	Goldstein (2014), Patel et al. (2023)	
		ampC	Firstly, the basic ampC β-lactase is produced, and then the antibiotic accumulates cell wall degradation products and competes with UDP-N-acetylpeptide for binding to AmpR. As the binding of UDP-N-acetylpeptide to AmpR decreases, AmpR undergoes conformational changes, leading to an increase in ampC production.	Tamma et al. (2019)	
	Tetracyclines (Tigecycline)	tet(X)	Covalent inactivation of all tetracyclines is achieved by the addition of hydroxyl groups at position C-11a, located between the C and B rings of the tetracycline core.	Grossman (2016)	
	PolymyxinE	mcr-1	The mcr-1 gene encodes a phosphoethanolamine transferase, which is responsible for modifying the lipid A portion of LPS by the addition of phosphoethanolamine, thereby decreasing its binding affinity for mucin.	Barlaam et al. (2019)	
Enterococcus faecalis	β-lactams	pbp4	Reduced affinity of penicillin for pbp4 occurs through the production of β-lactamases that can hydrolyze and inactivate the drug or because of point mutations in the penicillin-binding domain.	Conceição et al. (2014), Hiraga et al. (2008)	
	Oxazolidinones (Linezolid)	optrA	The optrA gene encodes the ABC-F protein, which generates resistance through ribosomal protection, specifically, active translocation of the antibiotic from its ribosomal target site.	He et al. (2016), Schwarz et al. (2021)	
	Tigecycline	tetM, tetO, tetS	Competitively binds to bacterial ribosomes and interferes with tetracycline-ribosome binding. Confers resistance by ribosome protection (RP).	Huys et al. (2004)	

Antibiotic resistance of B. fragilis

Studies have shown that the enterotoxin-producing type B. fragilis shows varying degrees of resistance to currently used antimicrobial drugs, such as clindamycin, cephalosporins, metronidazole, and carbapenem antibiotics (Yuran, Jiali & Jing, 2020). B. fragilis has a complex pattern of drug resistance. A study from Tehran, Iran, demonstrated a high level of resistance (95.8%) to ampicillin in B. fragilis isolated from biopsy specimens from patients with inflammatory bowel disease (Rashidan et al., 2018), while a study from Pakistan demonstrated a 20% resistance to metronidazole in B. fragilis isolated from clinical specimens (Shafquat et al., 2019). A study from Hebei, China, showed that drug sensitivity testing of isolated strains of B. fragilis showed 1.2% resistance to metronidazole, 6.1% resistance to chloramphenicol, and 22.0% resistance to carbapenems (Junhua et al., 2021). Metronidazole is one of the most commonly used antibiotics for the treatment of anaerobic infections, especially those of B. fragilis. Carriage of the nim gene is responsible for the development of resistance to metronidazole in this anaerobic bacterium as shown in Table 1.

Antibiotic resistance of F. nucleatum

The role of F. nucleatum in CRC is well established, and the use of effective antibiotics against F. nucleatum may be able to play a key role in ameliorating gut microbiota dysbiosis, thereby slowing the development of CRC, but inappropriate use of antibiotics can also affect the progression of CRC, and F. nucleatum is resistant to macrolide antibiotics, so these antibiotics are not recommended (Mihai et al., 2021). Carbapenems, β-lactam/β-lactamase inhibitor combinations, metronidazole, clindamycin, and moxifloxacin are used in clinical practice for the treatment of infections caused by F. nucleatum (Shilnikova & Dmitrieva, 2015). Studies have demonstrated that F. nucleatum exhibits a high degree of resistance to tetracycline, doxycycline, metronidazole, clindamycin and erythromycin (Ardila, Bedoya-García & González-Arroyave, 2023; Ardila & Vivares-Builes, 2022; Bullman et al., 2017).

Antibiotic resistance of E. coli

E. coli is strongly associated with the progression of CRC, and studies have shown that increased E. coli abundance is observed in CRC patients, not only is it a microbe that is already present in the gut, but the human body also receives E. coli colonization through drinking water, and studies have shown that drug-resistant strains of E. coli are already present in the water (Mihai et al., 2021). A study from Armenia showed that E. coli isolates from drinking water carried polymyxin resistance genes (Karpenko et al., 2024). A study from Zhejiang, China, showed that E. coli had a resistance rate of 85.0% to ampicillin, 55.1% to ciprofloxacin, 53.9% to ceftriaxone, 52.7% to cotrimoxazole, and 52.1% to levofloxacin (Zhen, 2018). Studies have shown that E. coli is resistant to the usual quinolone antibiotics (Fritzenwanker et al., 2018). This shows that E. coli is resistant to common antibiotics. Neonatal E. coli invasive isolates from developing countries have been reported to be up to 100% resistant to ampicillin and up to 90% resistant to gentamicin (Cole et al., 2019). In addition, the combined use of multiple antibiotics has led to the gradual development of multidrug resistance in E. coli, with the results of a study from Indonesia showing that 57.3% (47/82) of E. coli isolates were resistant to more than three antibiotics at the same time, with 39 patterns of resistance (Kallau et al., 2018).

Antibiotic resistance of E. faecalis

E. faecalis has been associated with the development of CRC tumors, and its excellent adaptability has led it to become a major contributor in the spread of antibiotic resistance. A study from Europe, including Germany, Norway, Denmark, Spain, Italy, Poland, France, Ireland, and Portugal, showed that the percentage of linezolid-resistant E. faecalis reported in each of these countries varies, but suggests that resistant strains have spread throughout Europe (Bender et al., 2018). Results of a study from Zhejiang Province, China, showed that among 74 clinical isolates of E. faecalis, the antibiotics with a high degree of resistance were mainly tetracycline and erythromycin, with a resistance rate of 89.2% and 73.0%, respectively; followed by quinolones ciprofloxacin and levofloxacin, with a resistance rate of 39.2% and 36.5%, respectively; once again, the high concentration of gentamicin has a resistance rate of 32.4% (Hong et al., 2019). Results of a study from Saudi Arabia suggest that most of the E. faecalis isolates had a multidrug resistance pattern (Farman et al., 2019).

Antibiotic resistance mechanisms in four intestinal bacteria

With the extensive use of antibiotics and long-term use, many bacteria have gradually developed different degrees of resistance to different antibiotics, and their resistance mechanisms have also become complicated. Exploring the mechanism of bacterial resistance can help to improve antibiotics and develop new drugs, and this review will elaborate on the resistance mechanisms of four CRC-associated gut microbiota.

Antibiotic resistance mechanisms in B. fragilis

Earlier studies have shown that antibiotic-resistant strains of B. fragilis can compromise the treatment of the disease, resulting in less effective therapies (Sherwood et al., 2011). Spigaglia et al. (2023) showed that strains of B. fragilis isolated from biopsy tissues of the colorectum generally showed resistance to ampicillin, cefoxitin and tetracycline. According to research reports, B. fragilis isolates have multiple determinants affecting drug resistance, such as multidrug efflux pumps, cfiA and nimB genes, and activating insertion sequences (Boyanova, Markovska & Mitov, 2019).

Drug resistance mediated by resistance genes

Research has shown that the nim gene encoding the Nim protein is present on plasmids or chromosomes (Haggoud et al., 1994). Regarding the mechanism of how the nim gene causes bacterial resistance to metronidazole, experts and scholars believe that the Nim protein, which has the properties of a nitroreductase enzyme, reduces metronidazole to a nontoxic aminopyrimidazole by transferring six of its own electrons to the nitro group of metronidazole, thus leading to the development of bacterial resistance to metronidazole (Fig. 3) (Deyan et al., 2023). Resistance to carbapenems in isolates of B. fragilis is mainly due to the presence of the cfiA gene, which encodes a metallo-β-lactamase (MBL) whose main mechanism of action is to inhibit β-lactam antibiotic activity by hydrolyzing the amide moiety of the β-lactam ring (Yekani et al., 2022), cfiA-positive strains usually show resistance to almost all β-lactam antibiotics (Wang et al., 2023). The results of the study showed that cfiA was present in all carbapenem-resistant isolates of B. fragilis (Gao et al., 2019). It has been shown that the cfiA gene is not highly expressed or may be ‘silent’ in most strains of B. fragilis, but is highly expressed when certain insertion sequence (IS) elements or some non-IS-mediated activation mechanism is mutated upstream of it, leading to high drug resistance in B. fragilis. The cfiA genes are not inactivated but only not expressed (Yekani et al., 2022). Among the β-lactam antibiotic resistance in B. fragilis mainly associated with the cepA gene and cfxA it carries, Niestępski et al. (2019) showed that 55 out of 123 (44.72%) BFG strains showed phenotypic resistance to ampicillin, and that 23 out of 55 (41.82%) resistant strains carried the β-lactam (cepA and cfxA) resistance genes. In addition, the researchers found the tetracycline resistance gene tetQ, macrolide and clindamycin resistance genes ermF in B. fragilis (Junhua et al., 2021). A genotyping study of clinical isolates of multidrug-resistant B. fragilis from India showed that these strains tended to express combinations of two or more resistance genes, e.g., two different resistance genes were coexisting in 25.8% of the strains, three different resistance genes were coexisting in 33.8% of the strains, and four different resistance genes were coexisting in 3.2% of the strains, with combinations of ermF and cepA being more common. The combination of ermF and cepA was more common, while cfiA, ermF and cepA were more frequently present in strains containing three resistance genes (Colney, Antony & Kanthaje, 2021). In summary, the major resistance genes in B. fragilis were nim, cfiA, cepA, cfxA, tetQ, and ermF; and the major genes that led to the development of multidrug resistance in B. fragilis were the simultaneous presence of cfiA, ermF, and cepA in the strain. Among them, both cfxA and cepA genes are major resistance genes for β-lactam antibiotic resistance, both belong to class A β-lactamases, and have some functional similarity, but they have different sequences, low homology, and may be weakly related genomically and genetically (Eitel et al., 2013).

Figure 3 Nim-mediated metronidazole resistance.

Horizontal gene transfer-mediated drug resistance

Acquisition of mobile genetic elements such as plasmids carrying drug resistance genes by bacteria through horizontal gene transfer (HGT) is one of the main ways for them to develop drug resistance (San Millan, 2018). Studies have shown that antibiotic resistance genes can be transferred horizontally by a variety of mechanisms, the most important of which are transformation, transduction, and conjugation (Fig. 4) (McInnes et al., 2020).

Figure 4 Mechanisms of horizontal gene transfer-mediated drug resistance.

Resistance gene transformation mainly refers to the uptake of naked DNA from the extracellular environment of B. fragilis by B. fragilis, which is then admixed into the host genome of the bacterium through homologous recombination (Johnston et al., 2014), As a result, B. fragilis has also acquired a corresponding antibiotic resistance. According to the study, the transformation mechanism is largely related to the genome of the recipient bacteria, and these genes or genomes are involved in exogenous DNA uptake and integration into the chromosome of the recipient bacteria (Michaelis & Grohmann, 2023). The discovery of bacterial membrane vesicles solved the mystery in the researchers’ minds, as free DNA is not stable and its time to remain intact outside the cell is short, which seems to limit the realization of transformation. The study suggests that the transport of drug-resistant genes between bacteria is also linked to membrane vesicles, and that certain drug-resistant genes or β-lactamases may transfer material by fusing with cells (de Sousa, Lourenço & Gordo, 2023). Membrane vesicles are spherical structures of 20–250 nm. Membrane vesicles enable the transmission of drug-resistant genes by fusing with target cells (McInnes et al., 2020). In vitro production of membrane vesicles containing β-lactamases by intrinsically drug-resistant B. fragilis, which then fuses with target cells that ingest the membrane vesicles to deliver the corresponding resistance genes (Stentz et al., 2015), This leads to the development of corresponding resistance in the cells of the target bacteria.

The mechanism of transduction of drug-resistant genes mainly refers to the transfer of drug-resistant genes between bacteria via phages, which play a central role in mediating the horizontal transfer of drug-resistant genes (Chiang, Penadés & Chen, 2019). According to reports, phage communities are widely present in the human gut (Shkoporov & Hill, 2019), which carry antibiotic resistance genes in large numbers (Debroas & Siguret, 2019). It was found that φB124-14 served as a human gut-specific phage whose original host was B. fragilis, and the phage was also known as a mobile macrogenome in the human gut because of its richness in multiple antibiotic resistance genes (Ogilvie et al., 2012). The abundance of these phages carrying antibiotic resistance genes increased significantly in the human gut after antibiotic treatment (McInnes et al., 2020).

The mechanism of conjugation of resistance genes mainly refers to the transfer of mobile genetic elements such as plasmids and integrative conjugative elements from one bacterium to another (McInnes et al., 2020). Conjugative transposons (CTn) are segments of DNA, mobile genetic elements integrated into chromosomes, which are able to move the relevant resistance genes to new locations precisely in the same or different DNA molecules in certain bacterial cells, resulting in the production of the corresponding drug resistance (Boiten et al., 2023; Partridge et al., 2018). It has been reported that the major resistance genes associated with B. fragilis, such as cepA for cephalosporins, ermF for MLSB analogs, and tetQ for tetracyclines, are essentially carried on the chromosome by conjugative transposons (Sóki et al., 2016). Among them, Boiten et al. (2023) showed that resistance to tetracycline in B. fragilis increased from an initial 20% to 80% in 20 years, and that the tetQ gene located on the conjugation transposon may be the underlying mechanism. It has been studied that transposons may also undergo mutations during the course of bacterial development, Tn5520 is a transposon that is mobile in B. fragilis. The Tn5520-like transposon in the isolate identified by Cao et al. (2022) belongs to two new variants (Tn6995 and Tn6996), which differ from the original Tn5220 in that they have ermF genes, which lead to resistance to streptozotocin. Studies have shown that antibiotic-stimulated B. fragilis enhances the secretion of membrane vesicles, which are enriched with immune stress factors, including proteins, nucleic acids, and peptidoglycans, thereby exacerbating the inflammatory response and cytokine secretion, which is one of the major factors in the development of CRC (Gilmore et al., 2022; Kozhakhmetova et al., 2024).

Overexpression of bacterial multidrug active efflux systems

Multidrug efflux pumps play an important role in the process of bacterial drug resistance, in which bacteria utilize efflux pumps to reduce the concentration of drugs in their own bodies and develop drug resistance (Huang et al., 2022). Existing studies have identified six drug efflux system superfamilies, namely, the major facilitator super family (MFS), small multidrug resistance (SMR), ATP binding cassette (ABC), and resistance nodulation and cell division (RND), multidrug and toxic compound extrusion family (MATE), and proteobacterial antimicrobial compound efflux family (PACE) (Zhouxing et al., 2022). Among them, the PACE family proteins have a relatively narrow drug-substrate recognition spectrum, which mainly includes some synthetic biocides such as chlorhexidine and acridine yellow, whereas the transporter proteins from the RND superfamily recognize a large number of different antibiotics and biocides (Hassan et al., 2018). According to research, certain efflux systems consist of a series of transporter proteins that remove a variety of foreign substrates from the bacterial cell, thus reducing the effects of multiple drugs on the bacteria, and may be referred to as multidrug efflux pumps (Huang et al., 2022). The emergence of multidrug efflux pumps has been reported to be one of the major causes of multidrug resistance in B. fragilis, and RND-type efflux pumps and MATE-type efflux pumps are prevalent in wild-type strains of B. fragilis (Ghotaslou, Yekani & Memar, 2018). A homology search of the B. fragilis genome identified 16 homologs of multidrug-resistant RND efflux pumps in the operon named bmeABC1-16, which consists of bmeA, bmeB, and bmeC, bmeB is an important antimicrobial efflux pump that is induced by exposure to a variety of antibiotics in the B. fragilis bmeB gene and the emergence of multi-drug resistant strains. BexA is present in many microorganisms and belongs to the MATE efflux family, leading to high levels of microbial resistance to antibiotics, and studies have shown that BexA is present in B. fragilis (Pumbwe et al., 2006; Ueda et al., 2005). Studies have shown that overexpression of multidrug efflux pumps is increasingly closely associated with bacterial drug resistance during clinical treatment of infections (Deyan et al., 2023), particularly for the emergence of multidrug-resistant B. fragilis: overexpression of the efflux pump plays an important role in the resistance of B. fragilis to antimicrobial agents such as β-lactams, fluoroquinolones, tetracyclines, fusidic acid, neomycin, metronidazole, and other virulence compounds, including triclosan, sodium dodecyl sulfate (SDS), and cholestrol salts (Ghotaslou, Yekani & Memar, 2018).

Mechanisms of antibiotic resistance in F. nucleatum

Mechanisms of β-lactamase-mediated antibiotic resistance

It was shown that the FUS-1 enzyme found in F. nucleatum is the first of its class D β-lactamase-producing enzymes (Dupin et al., 2015). Class D β-lactamase genes, often identified as intrinsic resistance determinants in environmental bacteria, occur in mobile genetic elements carried by clinically important pathogenic bacteria (Yoon & Jeong, 2021), Reported FUS-1 genotype of F. nucleatum from a clinical isolate of human pathogenic F. nucleatum (Voha et al., 2006).

Other resistance mechanisms

Studies have shown that exposure to a particular antibiotic interferes with the susceptibility of F. nucleatum to several antibiotics and may reduce susceptibility to antibiotics with similar mechanisms of action or the same resistance mechanism (de Souza Filho et al., 2012). de Souza Filho et al. (2012) showed that selected β-lactam strains were also much less susceptible to chloramphenicol and metronidazole. However, it has been shown that while resistance to β-lactam antibiotics in most Gram-negative bacteria is mediated by β-lactamase production, other mechanisms of antibiotic resistance include changes in penicillin-binding proteins, decreased permeability, or increased efflux pump activity (Huemer et al., 2020), Thus, in the study by de Souza Filho et al. (2012), it was again noted that no significant differences were observed in the antimicrobial drug susceptibility patterns of ampicillin and ampicillin-sulbactam, which suggests that cross-resistance between β-lactams, chloramphenicol, and metronidazole may indicate the induction of common mechanisms of resistance, such as changes in cell wall permeability. This suggests that there is a mechanistic association between antibiotics and that perhaps a combination of antibiotics may be effective in treating resistant strains. Studies have shown that antibiotic combinations can be more effective than the use of a single antibiotic because our bodies harbor a variety of microbiota, including aerobic, anaerobic, and facultative bacteria. Baumgartner & Xia (2003) demonstrated that combining metronidazole with penicillin V increased the activity of metronidazole (Baumgartner & Xia, 2003; Jacinto et al., 2008). A study from the United States showed that F. nucleatum utilizes type IV pili to facilitate the natural ability to import DNA and transfer genes horizontally, and found that F. nucleatum ATCC 23726 was able to ingest genomic DNA containing a chloramphenicol resistance gene and subsequently induced horizontal transfer of the gene to the chromosome of a wild-type strain (Sanders et al., 2023).

Antibiotic resistance mechanisms in E. coli

E. coli is one of the most common bacterial species in our intestinal system. A total of 60% of E. coli isolates isolated from biopsy samples of CRC patients were resistant to multiple antibiotics by Tariq et al. (2022). A study from Mexico showed that E. coli bacteria isolated from colorectal disease tissues were highly resistant to more than 10 antibiotics, including ampicillin, tetracycline, and ciprofloxacin (Canizalez-Roman et al., 2021). This suggests that there may be a greater correlation between CRC development and drug-resistant strains of E. coli.

Drug resistance mediated by mutations in drug resistance genes

First, it has been shown that mutations in drug-resistant genes are one of the main mechanisms for the development of drug resistance in E. coli, including spontaneous mutations, hypermutations, and adaptive mutations (Pulingam et al., 2022). Among them, Rodríguez-Verdugo, Gaut & Tenaillon (2013) showed that spontaneous mutations can be driven by, for example, interfering with DNA replication, and that resistance to rifampicin in E. coli is achieved by mutations in the rpoB gene encoding an RNA polymerase; hypermutation confers an evolutionary advantage to the bacterial species during adaptation to new environments or stressful conditions (Oliver & Mena, 2010); adaptive mutations refer to mutations at the transcriptional level that occur in the bacterial genome to adapt to changes in the survival environment, and when the environmental stress is lifted, the bacterial genome returns to its original condition (Fernández & Hancock, 2012; Pulingam et al., 2022).

According to the study, AmpR and AmpC are encoded by the ampR and ampC genes, respectively, suggesting that the ampR and ampC genes are related, and that AmpR serves as a transcriptional activator that binds to the cis-trans region upstream of the ampC gene promoter, thus acting to regulate AmpC (Philippon et al., 2022). Since the transcriptional regulator (AmpR) expressing AmpC is reportedly not present in E. coli, by what mechanism is the AmpC gene regulated? Haenni, Châtre & Madec (2014) showed that there are five highly conserved mutations in the promoter of AmpC, while the mutations in the attenuator are much more frequent, which are mainly attributed to spontaneous mutations in the promoter and attenuator (Haenni, Châtre & Madec, 2014; Kakoullis et al., 2021). This suggests to us that using promoter or attenuator mutation sites as drug targets may be an effective strategy to deal with AmpC-type E. coli drug resistance.

Hypermutagenic bacteria are microorganisms that have a stronger affinity for spontaneous mutations due to DNA repair defects or avoidance system errors (Oliver & Mena, 2010), resulting in greater adaptability to antibiotics. Hypermutagenic phenotypes of E. coli have been reported earlier (Denamur et al., 2002). According to studies, the mismatch repair system (MMR) is particularly important in the phenomenon of hypermutation, as it is not only one of the main causes of bacterial hypermutation, but also one of the main factors in the progression of CRC (Jin & Sinicrope, 2022). It has been shown that the most commonly mutated gene in strains with hypermutated phenotypes of E. coli is the mutHLS gene of the DNA methyl-orientation MMR pathway (Ellington et al., 2006).

Adaptive mutation denotes a temporary increase in the viability of a bacterium when it is attacked by an antibiotic, mainly due to changes in the bacterial genome or protein expression as a result of other environmental factors to which the bacterium is subjected, such as the nutrient conditions to which it is subjected or the sub-inhibitory concentration of the antibiotic itself (Fernández & Hancock, 2012). Simply put, adaptive mutations may be the induced mechanism by which bacteria produce genetic variability in a stressful state (McKenzie et al., 2000). It was shown that E. coli exposed to sublethal concentrations of streptomycin induced the expression of recA- and umuDC-independent mutant phenotypes on transfected M13 single-stranded DNA (Pulingam et al., 2022).

β-lactamase-mediated drug resistance

Another major mechanism by which E. coli develops resistance to β-lactam antibiotics is mediated by β-lactamase activity (Bush, 2018). The Ambler system based on sequence information indicates that β-lactamases are classified into four distinct classes called A, B, C, and D (Tooke et al., 2019). Epidemiologic investigations have shown that the prevalent strains have different rates and mechanisms of resistance at different times, in different regions, and in different populations (Xing, Bin & Lei, 2020). Strains with class A extended-spectrum β-lactamase genotypes (AmpA-type β-lactamases) are the most common, and class C β-lactamase (AmpC-type β-lactamases) genotypes are highly resistant, which has attracted extensive interest from researchers (Poirel et al., 2018).

Class A β-lactamases include penicillinase type 1 (PC1) (Tooke et al., 2019), TEM (named after Temoneira, the patient from whom the isolate originated) (Datta & Kontomichalou, 1965), sulfhydryl variant (SHV) (Chaves et al., 2001), Cefotaximase (CTX-M) (Bauernfeind, Schweighart & Grimm, 1990) and Klebsiella pneumoniae carbapenemases (KPC) (Rapp & Urban, 2012), it has been shown that the key to the ability of these class A β-lactamases to render antibiotics less effective is their ability to propagate on plasmids and other mobile genetic elements in a range of Gram-negative bacteria, as well as the fact that they broaden their spectrum of activity as new substrates are discovered in the clinic, which is also referred to in clinical terms as “extended-spectrum” phenotypic β-lactamases (ESBLs) (Tooke et al., 2019). The production of extended-spectrum β-lactamases is the main reason for the resistance of E. coli to β-lactam antibiotics. At present, the most reported extended spectrum β-lactamases are CTX-M type enzymes, which can be divided into five categories: CTX-M-1, CTX-M-2, CTX-M-8-8, CTX-M-9, and CTX-M-25 (Seo & Lee, 2021).

Finally, with the emergence of multidrug-resistant strains, some highly effective antibiotics have become “antibiotics of last resort”, with polymyxin E considered to be the last line of defense against multidrug-resistant and carbapenem-resistant Gram-negative bacteria, but in recent years there have been an increasing number of reports of colistin (polymyxin E) resistant bacteria (Hussein et al., 2021). Liu et al. (2016) revealed by whole plasmid sequencing that polymyxin E resistance may be caused by the plasmid-mediated mcr-1 gene. Dadashi et al. (2022) showed that the prevalence of colistin-resistant E. coli containing the mcr-1 resistance gene was reported to be 66.72%, 25.49%, 5.19%, 2.27%, and 0.32% in Asia, Europe, the Americas, Africa, and Oceania. Tigecycline and colistin are the last antibiotics against carbapenem-resistant bacteria, and it was found that a plasmid encoding the colistin resistance gene, mcr-1, and the tigecycline-resistance enzyme, tet (X6), existed in the same strain of E. coli, and that the presence of the two plasmids made E. coli co-resistant to these two classes of antibiotics (Xu et al., 2021b). Researchers predicted that the emergence of plasmids co-integrating mcr-1 and tet (X4) would pose a significant threat to humans, Lu et al. (2021) obtained seven evolutionary plasmids carrying mcr-1 and tet (X4) in vitro and further demonstrated that the plasmids could be inherited. Shafiq et al. (2022) detected a broadly resistant E. coli isolate co-carrying plasmid-mediated blaNDM-5 and tet (X4) genes.

Horizontal gene transfer-mediated drug resistance

Tigecycline is used as a broad-spectrum glycylcycline antibiotic for the treatment of E. coli infections, but tigecycline-resistant strains have emerged clinically. The mechanism of resistance involves flavin-dependent monooxygenase (tetX), and studies have shown that the emergence of tetX can increase resistance to tigecycline (Li et al., 2016). The main resistance mechanism is that tet (X3/X4) can directly inactivate tigecycline potency through hydroxylation of carbon 11a (Cui et al., 2020). As tet (X3/X4) is present on mobile plasmids, this leads to horizontal transfer of resistance across strains and species. Studies suggest that high levels of plasmid-mediated tigecycline resistance genes tet (X3) and tet (X4) emerged in 2019, which poses a significant threat to global public health (Li et al., 2023).

It was shown that conjugative plasmid-mediated horizontal gene transfer is the main mechanism mediating the spread of antibiotic resistance genes in E. coli, conjugation plasmids are the main vectors for spreading antibiotic resistance and are elements that can mediate their own transmission through conjugation (Mota-Bravo et al., 2023; Palomino et al., 2023). Minja, Shirima & Mshana (2021) showed that out of 51 blaCTX-M-15 positive donor isolates, 45 transferred the plasmid via conjugation. It has been shown that E. coli performs horizontal gene transfer mainly through DNA released by cell lysis, and that it can transfer DNA to different bacteria by secreting vesicles loaded with plasmid DNA into the environment (Cooper, Tsimring & Hasty, 2017; Maeusli et al., 2020). According to Kulkarni, Nagaraj & Jagannadham (2015) the presence of antibiotics and stress response caused by changes in the host environment lead to the formation of vesicles in E. coli strains, and vesicles isolated from E. coli protect the bacteria from membrane-active antibiotics (colistin and polymyxin B), suggesting that vesicles protect the bacteria from the growth inhibitory effects of certain antibiotics.

Drug resistance in E. coli mediated by efflux pumps

As early as the 1990s, the drug-resistant nodular differentiation family (RND) was identified in E. coli and is represented by the AcrAB-TolC pump in E. coli (Fig. 5), which mediates bacterial multidrug resistance. RND is located in the inner membrane, and as a transporter protein, it must interact with periplasmic bridging proteins and outer membrane channels in order to excrete drugs directly across the inner, periplasmic, and outer membranes to the outside of the cell membrane (Li, Plésiat & Nikaido, 2015).

Figure 5 Mechanism of E. coli AcrAB-TolC-mediated efflux.

In the presence of E. coli AcrAB-TolC, drug efflux through the cell membrane forms an effective permeability barrier due to the presence of low-permeability pore proteins (i.e., “slow pore proteins”), which are capable of generating multidrug resistance (Li, Plésiat & Nikaido, 2015). It was shown that E. coli AcrAB-TolC is regulated by the multiple antibiotic resistance manipulator Mar, which is expressed as two separate transcriptional units, one of which is MarRAB, controlled by MarO, which specifies a Mar repressor (MarR), an activator (MarA) and a small protein (MarB), who are respectively encoded by marR, marA and marB, with MarB located downstream of MarA (Alekshun & Levy, 1999; Weston et al., 2018). Under normal conditions, MarR represses the MarRAB operon by binding to the two palindromic sequences of marO, but when antibiotics are encountered, repression of MarR is disrupted and transcription of marRAB occurs. It has been shown that de-repression of the marRAB operon results in the expression of MarA: each regulator promoter has a binding site called a “marbox” binding site, MarA undergoes positive feedback when it binds to DNA sequences upstream of the marbox, the repressor site of MarR, so this represses marR and allows marA to be activated. MarA expression promotes the activation of several genes in its regulator, including the AcrAB and TolC genes, which increases drug efflux and lead to multidrug resistance (Martin et al., 1999; Weston et al., 2018). It was shown that drug resistance aspects were affected in strains lacking the gene encoding the AcrAB-TolC multidrug efflux pump (ΔtolC or ΔacrB) (Kobylka et al., 2020).

Mechanisms of antibiotic resistance in E. faecalis

Drug resistance mediated by mutations in drug resistance genes

Resistance to β-lactam antibiotics in E. faecalis occurs by two mechanisms: production of β-lactamases and alteration of the affinity of penicillin-binding proteins for β-lactam antibiotics (Ono, Muratani & Matsumoto, 2005). Studies have shown that resistance to β-lactam antibiotics in E. faecalis can be mediated by the production of a non-inducible β-lactamase (Herrera-Hidalgo et al., 2023). The presence of penicillin-binding protein 4 (PBP4) in E. faecalis results in a low affinity for β-lactam antibiotics, which leads to a certain degree of resistance to β-lactam antibiotics in E. faecalis (Urban-Chmiel et al., 2022), and PBP4 is considered to be the key molecular basis for the resistance of E. faecalis to β-lactam antibiotics (Lazzaro et al., 2021). Epidemiologic data suggest that the progressive increase in resistance to β-lactam antibiotics in E. faecalis is attributable to overexpression of PBP4 (Lazzaro et al., 2021). PBP4 belongs to the class of transpeptidases involved in the formation of the peptidoglycan layer; whereas β-lactam antibiotics block peptidoglycan biosynthesis via PBP4 acylation (Moon et al., 2018; Timmler et al., 2022). Further studies revealed that PBP4-mediated resistance to β-lactam antibiotics in E. faecalis was associated with the CroRS two-component signaling system (TCS) (Kellogg et al., 2017). Timmler et al. (2022) showed a correlation between PBP4 and the CroR system, the exact mechanism of which requires further experimental confirmation.

Vancomycin is commonly used for severe drug-resistant Gram-positive bacterial infections (National Institute of Diabetes and Digestive and Kidney Diseases, 2012a), and plays a twofold role in the adjuvant treatment of CRC: on the one hand, vancomycin depletes butyrate-producing bacteria in the gut, thereby enhancing the efficacy of radiotherapy; on the other hand, it inhibits the bacteria that convert primary bile acids into secondary bile acids, thereby enhancing the efficacy of anticancer therapy (Singh et al., 2020; Yang et al., 2023). The ability of E. faecalis to cause the development of CRC has been certified, but recent studies have found that vancomycin-resistant E. faecalis plays a more pronounced role in the proliferation, migration, and angiogenesis of CRC cells compared to E. faecalis (Zhang et al., 2024a). Since the first discovery of vancomycin-resistant E. faecalis clone sequence type 796 (ST796) in Australia in 2011, drug-resistant strains are now widely reported worldwide (Li, Walker & De Oliveira, 2022). A total of nine vancomycin resistance cluster genes of the Van family have been identified, with VanA and VanB being the most common among clinical isolates (Raza et al., 2018; Zalipour, Esfahani & Havaei, 2019). Taji et al. (2019) showed that vanA gene was detected in 37.7% of E. faecalis isolates. Strateva et al. (2019) tested and characterized an isolated strain of E. faecalis and found that the vanA gene cluster was on a segregating overlapping cluster with two repetitive IS1216E sequences around its flanks, followed by transfer experiments by filter mating assay using E. faecalis JH2-2 as a receptor strain, which showed unsuccessful results in terms of transferring vancomycin resistance, suggesting that the possible location of the vanA gene cluster at the chromosomal position. According to the study, the Tn1549 transposon carries the vanB operon on it (Simar et al., 2023). Although there are fewer reports on VanB, it has attracted much attention from scholars because of its high detection rate (Sadowy, 2021).

The main mechanism of resistance to linezolid in enterococci involves the G2576T mutation in the 23S rRNA gene (Rodríguez-Noriega et al., 2020), and other mechanisms are mutations in the L3 and L4 ribosomal proteins as well as in two plasmid vector genes (cfr and optrA) (Arias & Murray, 2012). According to research, optrA is located in a new gene cluster containing the chloramphenicol output gene fexA. The protein product of optrA belongs to the ATP binding cassette (ABC)—F protein superfamily, and its resistance is mediated by ribosome protection. Compared with other gene determinants such as cfr or 23S rRNA and ribosomal protein mutations, mutations in optrA are a common cause of oxazolidinone resistance in E. faecalis (Roy et al., 2020). The research results of Deshpande et al. (2018) showed that isolates with chromosome localization of optrA exhibited different array structures. The flank regions of the optrA arrays of E. faecalis from Thailand and isolates from France were different. From the results of gene array analysis, it can be seen that with the continuous spread of optrA, a large degree of gene rearrangement is occurring, and the core genetic elements remain similar. However, in isolates from different geographical locations, their positions in the array are not the same (Deshpande et al., 2018). Therefore, the monitoring of the flanking regions of the optrA array is of great clinical importance.

Horizontal gene transfer-mediated drug resistance

The genome of E. faecalis was found to be highly plastic, and resistance to other antibiotics, such as high levels of aminoglycoside resistance, high levels of ampicillin resistance, and vancomycin resistance, is readily acquired through mutations in resistance genes or through horizontal transfer of genetic elements conferring resistance determinants (García-Solache & Rice, 2019). It was confirmed that transposons constitute the majority of the mobile genetic elements present in the genome of E. faecalis, and that Tn916, as the first confirmed conjugative transposon, carries tetracycline resistance and is able to be transferred to the chromosome of the recipient cell or to a conjugative plasmid by transposition, and that transposon incorporation into the conjugative plasmid increases the frequency of transfer (García-Solache & Rice, 2019).

Data from the China Antimicrobial Drug Surveillance Network (CDSN) showed that the high-level gentamicin resistance rate in E. faecalis ranged from 28.8% to 61.4% from 2005 to 2017, and was mainly mediated by the bifunctional enzyme encoded by the fused aac(6′)-aph(2″) gene in E. faecalis, 6′-acetyltransferase-2″-phosphotransferase (Ferretti, Gilmore & Courvalin, 1986). The aac(6′)-aph(2″) gene is plasmid-borne in most cases and is located on the E. faecalis Tn5281-like transposon (Zhang et al., 2018). The non-truncated form of Tn5281 consists of a central region containing the aac(6′)-aph(2″) gene flanked by inserted inverted repeats of sequence IS256, whereas the truncated form is the aac(6′)-aph(2″) 3′- or 5′-end, or both lacking IS256 (Zhang et al., 2018). Daikos et al. (2003) showed that 24 out of 30 isolates containing the truncated form transferred gentamicin resistance, while only 3 out of 34 isolates containing the nontruncated form transferred gentamicin resistance, suggesting that the truncated variant is mobile and more effective in transferring gentamicin resistance.

It has been shown that the bacterial type IV secretion systems (T4SSs) are a functionally diverse translocation superfamily, and that one of its major functional subfamilies is the conjugation system that mediates DNA transfer between bacteria, and that the conjugation system can propagate mobile genetic elements that typically encode bacterial resistance to antibiotics (Costa et al., 2021). The transferable plasmids of multidrug-resistant E. faecalis are T4SS with a functional plasmid-encoded (PE-T4SS) and a chromosome-encoded T4SS (CE-T4SS); compared with PE-T4SS, CE-T4SS exhibits different characteristics in protein structure and can mediate large genome-wide gene transfer (Hua et al., 2022). The study by Hua et al. (2022) identified a widely distributed CE-T4SS in E. faecalis, and to better understand the process of gene transfer, the researchers analyzed the oriT element (Hua et al., 2022). At the initiation site of horizontal gene transfer, the researchers identified four putative oriT with reverse complementary structural domains, oriT1-oriT4, and hypothesized experimentally that oriT4 is the required initiation site for horizontal gene transfer mediated by CE-T4SS in D5165. The investigators selected CE-T4SS+ reference strain ATCC 19433 as a donor and a popular erythromycin-resistant ST179 strain, S6008, as a recipient, suggesting that CE-T4SS induces gene transfer in the host (Hua et al., 2022).

While the previous discussion was about horizontal transfer of drug resistance genes in Gram-negative bacteria through membrane vesicles, we learned some results about whether membrane vesicles can work in Gram-positive strains of E. faecalis, which is a Gram-positive bacterium. Initial studies on whether membrane vesicles produced by E. faecalis carry out the transfer of drug-resistant genes showed that there was no detection that E. faecalis could carry out horizontal gene transfer via membrane vesicles (Afonina et al., 2021). However, a recent study by Zhao et al. (2024) showed that a strain of E. faecalis carrying a linezolid resistance gene could successfully transfer the resistance gene to another strain of E. faecalis via a membrane vesicle, and that the recipient strain still had the potential to transmit the resistance gene.

Multidrug efflux system-mediated drug resistance

EfrAB, a heterodimeric multidrug ATP-binding cassette (ABC efflux system family) transporter protein, causes endogenous resistance to antibiotics including fluoroquinolones in Enterococcus spp. (Shiadeh et al., 2020). Shiadeh et al. (2020) showed that ciprofloxacin-resistant E. faecalis isolates showed varying degrees of overexpression of efrA and efrB genes. A study by Esfahani et al. (2020) found no significant relationship between the upregulation of the expression of the efflux pump and the level of minimal inhibitory concentration, and the researchers found isolates without any mutations in the expression of efflux genes but with drug resistance, and furthermore, 23 homologs of the ABC family of transporter proteins were detected in E. faecalis isolates. The above evidence suggests that the development of fluoroquinolone resistance may be the result of ABC family transporter proteins but not necessarily EfrA or EfrB.

Tetracycline is one of the most commonly used broad-spectrum antibiotics, and many bacteria have developed resistance to this antibiotic, the most common mechanism involves membrane-associated proteins (TetA), which exclude the antibiotic from the bacterial cell before inhibiting peptide elongation (Ramos et al., 2005). According to research reports, tetracycline repressor protein (TetR) family proteins affect the tolerance of E. faecalis to tetracycline by controlling the expression of tetracycline efflux pump genes, such as the regulation of efflux pumps by the tetA gene (Gu et al., 2020). TetR proteins control the expression of the tet gene, the product of which confers bacterial resistance to tetracycline (Ramos et al., 2005). Research has shown that tetR and tetA are adjacent and oriented differently, and their gene products tightly control the expression of tetA and tetR (Ramos et al., 2005), TetR homodimers block the promoter of efflux pump genes by binding to repeat palindromic sequences in the upstream gene intergenic region using helix α1 to α3. In the presence of tetracycline, TetR homodimers interact with tetracycline and magnesium to form protein ligand complexes, which cause conformational changes in the TetR ligand complex, leading to the release of TetR homodimers from the promoter of efflux pump genes. This separation activates the expression of tetracycline related efflux pumps and squeezes tetracycline out of bacterial cells (Gu et al., 2020).

Commonalities and differences in drug resistance

Consistency of resistance mechanisms of the same bacteria to different types of antibiotics

Studies have shown that the same bacteria can be resistant to multiple antibiotics at the same time; are the resistance mechanisms similar? One of the main mechanisms for the development of resistance to carbapenems and metronidazole in B. fragilis is efflux pump-mediated resistance (Ghotaslou et al., 2018; Yekani et al., 2022), it has been shown that some efflux pumps are not substrate specific, and thus bacteria can excrete many chemically inconsistent drug substrates via efflux pumps, thereby increasing the emergence of multidrug resistance (Willers et al., 2017). The emergence of multidrug-resistant strains of bacteria has caused adverse effects on human health and society, against multidrug efflux pumps there are now a number of targeted efflux pump inhibitors, these inhibitors antibiotics can be used in combination to effectively minimize the adverse effects of multidrug efflux pump-mediated resistance mechanisms. Currently, effective efflux pump inhibitors such as reserpine and verapamil have been identified, which can enhance the antibiotic’s antimicrobial capacity, but due to the toxicity of these inhibitors is too strong and the clinical relevance of these inhibitors is not high, and has not been specifically used, the future of this field of research and development still need to be combined with the clinical (Lamut et al., 2019). According to reports, MBX2319, as an effective efflux pump inhibitor of the RND efflux system in E. coli, can inhibit ciprofloxacin, Levofloxacin and piperacillin reduced the MICs of E. coli strains by two fold, four fold, and eight fold (Zhang et al., 2024b).

The resistance of F. nucleatum to β-lactam antibiotics and chloramphenicol is mediated by the class D β-lactamase FUS-1 (Dupin et al., 2015) and the acetyltransferase (CAT) (de Souza Filho et al., 2012), respectively, which are the common mechanisms of resistance in this group of bacteria. The production of hydrolytic enzymes leads to bacterial resistance to many antibiotics, and experimental studies are currently underway in the clinic regarding the inhibition of active drugs that inhibit the production of hydrolytic enzymes as a type of hydrolytic enzyme inhibitor that can be used in conjunction with antibiotics under special circumstances, certain drugs with good efficacy and low toxicity (Rodríguez-Baño et al., 2018). At present, research on hydrolytic enzyme inhibitors mainly focuses on β-lactamases inhibitors, which represent the main strategy to combat the strong and widely spread resistance mediated by β-lactamases, examples of antibiotic combination therapy include amoxicillin clavulanic acid and piperacillin tazobactam, but these belong to class A β-lactamases inhibitors and have poor effects on Class D β-lactamases, and so far, bicyclic borates are inhibitors that can effectively act on class D β-lactamases (Tooke et al., 2019). However, there is still a need for increased clinical research on class D β-lactamase inhibitors because class D β-lactamases are understudied in many ways, there is great variability in sensitivity to inhibitors, and even mutations in resistance genes may have led to bacterial resistance to inhibitors (Evans & Amyes, 2014; Tooke et al., 2019). As early as the 1990s, it was noted that there is a carboxylesterase enzyme in mammals, mainly found in serum and intestinal mucosal sites, and that this enzyme converts diacetylchloramphenicol (a product of acetyltransferase) to active chloramphenicol, and although this is a low-level activity in humans, it suggests ways and means by which we can reduce antibiotic resistance by studying the products that are present in the human body (Sohaskey & Barbour, 1999).

The resistance of E. coli to rifampicin and β-lactam antibiotics is due to mutations in the rpoB and ampC genes, respectively (Kakoullis et al., 2021), it can be seen that mutations in resistance genes are the main resistance mechanism of E. coli. In previous studies, E. faecalis has developed varying degrees of resistance to β-lactam antibiotics, linezolid, and vancomycin. The resistance mechanisms are attributed to overexpression of the β-lactam antibiotic binding protein PBP4 (Lazzaro et al., 2021), the G2576T mutation in the 23S rRNA gene and mutations in the L3 and L4 ribosomal proteins as well as in two plasmid-borne genes (cfr and optrA) (Rodríguez-Noriega et al., 2020), and the vanA gene cluster located at chromosomal location in E. faecalis mediated vancomycin resistance (Strateva et al., 2019), which shows that resistance gene mutations are a common resistance mechanism in E. faecalis during the development of resistance to different antibiotics. There is a growing interest in the treatment of antibiotic resistance mutations, and different therapeutic techniques and combination therapies are being developed.

Different bacteria have different resistance mechanisms to the same antibiotics

Are there similarities in the resistance mechanisms of different bacteria facing the same antibiotic? Both E. coli and B. fragilis are resistant to carbapenem antibiotics, and E. coli resistance to carbapenem antibiotics is due to carbapenemase production (Nordmann, Naas & Poirel, 2011); whereas B. fragilis resistance to carbapenem antibiotics is mainly due to increased expression of the cfiA gene (Yekani et al., 2022). The increase in drug resistance of B. fragilis is mainly due to the horizontal transfer of tetracycline resistance genes (Boiten et al., 2023); whereas, for E. coli, flavin-dependent monooxygenase (tetX) leads to an increase in tigecycline resistance (Li et al., 2016). This shows that different bacteria are resistant to the same antibiotic, but they have different mechanisms of resistance. In addition, the main mechanism of cephalosporin resistance in B. fragilis is the expression of β-lactamase encoded by the cepA gene (Jasemi et al., 2021), in contrast, the expression of low-affinity penicillin-binding proteins in E. faecalis is responsible for the development of resistance to most cephalosporins in E. faecalis (Manoharadas et al., 2023).

Summary and outlook

The cause of CRC development is unknown, people with the highest antibiotic exposure have the greatest risk of CRC compared to those who use the least amount of antibiotics, and repeated use of antibiotics can lead to abnormal infections of gut microbiota as well as an increase in drug resistance, which may increase the risk of CRC (Aneke-Nash et al., 2021). Previous studies have shown that antibiotic use and the development of CRC usually show a positive correlation between the two (Cao et al., 2018), the inappropriate use of antibiotics, such as the duration of the medication as well as the size of the dose, leads to drug resistance. However, most of the current studies have limited information or do not have enough certainty to perform extensive analysis on the type, dose or duration of antibiotics, as well as tumor stage and site, and still need to be followed up by researchers (Lu et al., 2022). In this review, we describe in detail the resistance mechanisms of four CRC-associated gut microbiota, including horizontal gene transfer, changes in the expression of active efflux mechanisms, and changes in the expression of their unique resistance genes. It can be seen that the four bacteria have different mechanisms of drug resistance, but in general, the effect of drugs on bacteria and resistance is not a simple “behavior”, but a systematic and complex linkage change, according to our summary of the resistance mechanisms of the four bacteria, it can be seen that the resistance mechanisms of the four bacteria have their unique aspects, therefore that better utilization and management of these antimicrobials is a long way to go.

We found a deeper connection between the antibiotic resistance mechanisms of these four gut microbiota. Firstly, it was discovered that bacteria have different resistance mechanisms to the same antibiotic, which may be related to the antibacterial mechanism of antibiotics against bacteria. Therefore, starting from the bacteria themselves, antibiotics can be used to target a certain structure of the bacteria, such as the cell wall or membrane. Nanotechnology has achieved significant results in treating antibiotic resistance in the past decade. Some nanoparticles can prevent or overcome the formation of bacterial biofilms, thereby enhancing antibacterial effects. We have also explored the same resistance mechanism from the perspective of homologous bacteria when they are confronted with different antibiotics, which was seen in all four CRC-associated bacteria, suggesting a generalized resistance mechanism to different types of antibiotics, which may lead to the development of an aspect of the bacteria that the antibiotic is not able to kill, leading to the widespread spread of resistant strains and multidrug resistance. Then for some antibiotics whose resistance has spread widely, there is a need to work on drug improvement and research and development. A wave of novel drugs has emerged against ultra-broad-spectrum β-lactamase-producing E. coli, and LysSAP-26 is a promising active drug, which has been shown to be effective in inhibiting the growth of both E. coli and E. faecalis in vitro (Lim et al., 2024). Second, modulation of RNA expression in bacteria can effectively control the development of antibiotic resistance. RNA attenuators in the mRNA leading region combine the expression of resistance genes with the action of antibiotics, and compounds have been developed that can modulate bacterial resistance by modulating RNA and its attenuators (Dersch et al., 2017). Currently, drug combination therapies have been effective in addressing the current status of drug resistance in B. fragilis, and imipenem and minocycline have been shown to have a synergistic antimicrobial effect in B. fragilis (Umemura et al., 2022). In conclusion, the development of CRC is critically linked to the emergence and growth of drug-resistant bacteria, and the expansion of drug-resistant bacteria cannot avoid the large-scale and high-dose use of antibiotics, which leads to gut microbiota dysbiosis and subsequent carcinogenesis. This suggests that modern medicine still needs to make efforts in the development and improvement of antibiotic.

Additional Information and Declarations

Competing Interests

The authors declare that they have no competing interests.

Author Contributions

Yu Gan conceived and designed the experiments, performed the experiments, analyzed the data, prepared figures and/or tables, authored or reviewed drafts of the article, and approved the final draft.

Hao Yang conceived and designed the experiments, analyzed the data, authored or reviewed drafts of the article, and approved the final draft.

Maijian Wang conceived and designed the experiments, analyzed the data, authored or reviewed drafts of the article, and approved the final draft.

Jida Li conceived and designed the experiments, performed the experiments, prepared figures and/or tables, and approved the final draft.

Data Availability

The following information was supplied regarding data availability:

This is a literature review.

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
