# Peer review of "Advances in drug resistance and resistance mechanisms of four colorectal cancer-associated gut microbiota"

_PeerJ, doi:10.7717/peerj.19535_

## Round 0.1 · original submission · Major Revisions

The Authors are strongly recommended to fully and clearly the comments raised by the Reviewers 1 and 2. In particular, please also pay particular attention to the comments displayed by the Reviewer 3.

·

Basic reporting

This review offers relevant information on the mechanisms of antibiotic resistance in 4 species that are highly linked to colorectal cancer. Overall, it is a good manuscript, but it needs to be improved and expanded, especially to make the relationship between these mechanisms of antibiotic resistance and the development of colorectal cancer explicit and clear.

Try not to mix results in humans with results in a murine model.

Attached to this review is a document with various notes you can use to improve your work.

Experimental design

The design of this study is correct and replicable. However, the authors are not taking full advantage of the information collected in the literature search.

Please review the comments in the attached document.

Validity of the findings

The conclusions section is extensive. The authors should reduce it while expanding the discussion section so that we can adequately discuss the results and draw solid, concrete conclusions based on the discussion developed within the manuscript.

Reviewer 2 ·

Basic reporting

The manuscript of Gan et al. is review on the antibiotic resistance mehanisms of four main CRC driver organisms. Overally the information gathered is OK but it is not exhaustive and uneven very much. Additionally, the standard of communication is poor at many places.

My other comments are:
1. Since there are other articles summarizing more the antibiotic resistance mechanisms of each of these bacteria this review should add some specificity to this combination of CRC pathogens, indeed there is lilltle information on their pathogenic mechanisms and possible therapies.
2. Spaces are missing at many places before parentheses, e.g. lines 100, 107.
3. If you mean mice it would be better to use Min+, or Apc/Min+.
4. If you mention the main species the typing should be italic, e.g. lines 133, 142, 148, 159 etc.
5. Lanes 139-141: What is Mimicronium fragilis? Should have been expounded, and species names in italic. Nim genes are one causative agents of metronidazole resistance among Bacteroides, and gene names are also italic.
6. Lane 177: The abbreviated form should be used: B. fragilis.
7. Lane 181: On the other hand the names of proteins start with capital letters, so Nim proteins.
8. Lane 193: The proper expression is silence that may be in quotation marks like: and may be 'silence', but... The cfiA genes are not inactivated but only not expressed.
9. Lane 224: What vesicles, it should be specified.
10. Lane 251: B. fragilis with space.
11. Lane 346: recA and umuDC in italic.
12. Lanes 402-403: Unitntelligigble sentence, what does splicing in this case mean?
13. Lanes 467 and 471: In IS1216E and Tn1549 the numbers should be in italic according to widepread use and nomenclature. Additionally, vanB in italic also.
14. Lanes 502-503: Gene name latters in italic.
15. Lanes 530 and 532: Italicization again.
16. Lanes 559-561: the sentence is a factual error.
17. Lane 635: B. fragilis.
18. Lane 180: Problem in the title.
19. Figure 3 is too general. It is not needed.

Experimental design

Although the authors used many search terms the results are uneven. The numbers of found items to each term and the selection criteria to be used in the text and in the references should have been given.
Lots of articles examined the resistance rates of these organisms but it is hadly covered (lines 115-169). Few lines are not enough to cover these issues. The B. fragilis part discusses insignificant issues thus can be misleading.
The subsection levels in the manuscript should have been indicated somehow.

Validity of the findings

No comment.

Reviewer 3 ·

Basic reporting

No comment

Experimental design

No comment.

Validity of the findings

No comment

Additional comments

Advances in drug resistance and resistance mechanisms of four colorectal cancer-associated enteric bacteria

Gan et al. 2024

Summary

This review describes drug resistance in four colorectal cancer associated enteric bacteria, B. fragilis, F. nucleatum, E. coli and E. faecalis. It also describes the mobile genetic elements associated with each resistance gene and compares drug resistance between different organisms.

Feedback

While a good attempt has been made to consolidate current knowledge about antibiotic resistance in these pathogens, the review fails to give any greater detail nor insight than other reviews focusing on each organism individually. The introduction feels repetitive as the authors state the cases and mortality of colorectal cancer four times in a row. There are many occasions in the manuscript where sentences don’t make any sense e.g L. 181 “Protein is a determinant of metronidazole resistance”. The author’s review is framed from a colorectal cancer standpoint but they make no attempt to review resistance in organisms specifically isolated from colorectal tumours. The review is structured in a strange way with extremely short paragraphs that are made redundant by the following more detailed paragraphs. The authors have the habit of stating the results of certain studies as if this applies to all incidences of the organism. Unless the study is on a global or national level you cannot draw conclusions about the percentage of resistant organisms. The summary and outlook section could be used to make more specific recommendations and to offer an opinion on the best way to tackle these drug-resistant organisms in the context of colorectal cancer. I feel this review would be of limited interest to the wider field as it simply repeats what has been stated in other reviews.

---

## Round 0.2 · Major Revisions

There are some compulsory comments to be carefully addressed, raised by this Reviewer, which will improve the readability of your manuscript.

·

Basic reporting

The authors have addressed all my comments and suggestions well. This manuscript only needs profound proofreading to be suitable for publication at PeerJ.

Experimental design

The authors have addressed all my comments and suggestions well. This manuscript only needs profound proofreading to be suitable for publication at PeerJ.

Validity of the findings

The authors have addressed all my comments and suggestions well. This manuscript only needs profound proofreading to be suitable for publication at PeerJ.

Reviewer 3 ·

Basic reporting

I have read the revised version of the manuscript thoroughly. While there have been improvements, I feel that the authors have not addressed all the points raised in the first round of reviews. I have included below a detailed list of outstanding issues that should be addressed.

Ln26: Please include the word “of” between etiology and colorectal.
Ln26: I think you need to include a statement like “thought to be” before “one of the main factors”. It doesn’t really make sense to say the etiology is unknown and then state what one of the main factors is.
Ln33: It should be “abuse of antibiotics has made the problem of drug” not “abuse of antibiotics have made the problem of drug”.
Ln34: “Thorny” is not used in this manner and reads informally, consider changing it to another word. Please change all instances of “thorny” in this manuscript to a more suitable word.
Ln50: “involve” should be “involves”.
Ln62: I believe one of the other reviewers asked you to change “intestinal flora” to “gut microbiota”, please can you make sure you have done this throughout the manuscript.
Ln83: You say that there are few reports on drug resistance in the four CRC associated bacteria. This is not true, there are many papers that have discussed drug resistance in these organisms. Please remove this statement.
Ln104: You have written “Bacteriophage fragilis”, I assume that this is just a mistake in your manuscript and that you didn’t use this for your search. If you used this in your search you will likely have missed much of the literature associated with Bacteroides fragilis.
Ln137: Not all pks+ E. coli are pathogenic E. coli Nissle contains the pks island and has been used a probiotic for many years. Please alter your text to reflect this fact.
Ln140: E. coli does not produce colistin. It produces the genotoxin colibactin. This is very sloppy.
Ln171: As far as I am aware antibiotics are still the first choice for treating “harmful bacteria”. Please clarify this statement.
Ln173: The antibiotics are not related to colorectal cancer, they are used to treat bacteria which have been associated with colorectal cancer. Please rewrite this statement.
Ln193-194: You must be specific when you are talking about beta-lactams as this sentence doesn’t make sense when you talk generally about “beta-lactam antibiotics”.
Ln195: You can’t say low globally and “generally <90%”, this could mean 89% which can in no way be considered low.
Ln197: As I stated in my last review, you cannot use the results from a study conducted in one hospital or country to make a sweeping generalisation about drug resistance in a whole species. If you are going to use these papers you must be specific and state where the study was conducted.
Ln214: You need to be specific, is it the carriage of E. coli generally, a specific strain, or the abundance of E. coli that is related to the progression of CRC?
Ln214: “microbiota” is not the correct word it should be something like “microbe” or “bacterium”.
Ln219: I have the same problem with this section as I stated in Ln197. You need to go through your entire manuscript and be more specific with your writing.
Ln226-228: This sentence is unintelligible, please re-write so that it makes sense.
Ln230: Enterococcus faecalis should be in italics. You were told in the previous reviews to check your use of italics with bacterial names, genes names etc. Please read through your entire manuscripts and make sure that you have done this correctly!
Ln231: The use of the word “microbiota” is incorrect. Please change it to something like “contributor”.
Ln232: What do you mean by “the highest resistance to linezolid”? Please clarify this statement.
Ln241: This study doesn’t add anything to the paper as it’s an isolated study that almost certainly doesn’t have relevance to global E. faecalis resistance rates.
Ln270: ‘silence’ should be ‘silent’.
Ln272: Please could you clarify what you mean when you say “The cfiA genes are not inactivated but only not expressed.”?
Ln287: “multimandibular” should be “multidrug”.
Ln296: “heaviest” is not an appropriate word. “most important” would be better.
Ln380: “removable” is not the correct word. This should be changed to “mobile” throughout the manuscript.
Ln402: Did you forget to include a HGT section for F. nucleatum?
Ln470: This line doesn’t make sense.
Ln473-476: These lines repeat themselves. Please condense them.
Ln491: Modify the line to read something like “Shafiq M et al. detected a broadly resistant E. coli…”
Ln509: “Splicing” is not an appropriate word. Please use the correct word (conjugation, transformation or transduction.)
Ln515: Can you clarify why you have included “mucins”? Mucin is not an antibiotic.
Ln522: Please finish this line “external…”.
Ln529: “Deterrent” isn’t an appropriate word, please use “repressor” instead.
Ln530: Again, please make sure that you are italicising gene names!
Ln534: I’m not sure what you mean by “manipulator”, but it’s not correct. I think you mean the “marRAB operon”.
Ln541: Again, genes names not italicised. I’m going to stop pointing this out, but you need to read your manuscript carefully and italicise where appropriate.
Ln546-547: This line is wrong. Beta-lactamases don’t mediate the overproduction of penicillin-binding proteins. Please re-read these lines and clarify what you mean.
Ln577: As with Ln509, “splicing” doesn’t make any sense. “Transfer” would work better.
Ln581: “manipulator” doesn't make any sense. I think you mean “operon”.
Ln587: It should be “optrA”.
Ln606: IS and Tn names should be written with the number italicised. Please read through your manuscript and change it where appropriate.
Ln623: Again, “splicing” is the wrong word.
Ln625: “removable” should be changed to “mobile”.
Ln631: “oriT” should be italicised.
Ln687: What is “Jung’s drug resistance mechanism”?
Ln692: You need to finish the line. “clinical use”?
Ln695: What do “proteases” have to do with this?
Ln701 & Ln703: “beta-lactase” should be “beta-lactamase”.
Ln707: “thrown” does not make sense in the sentence and should be deleted.
Ln711: Remove “no” from the sentence, it doesn’t make sense.
Ln713: “Movement” doesn’t make sense. Do you mean effect?
Ln728-739: While this text is interesting, it has nothing to do with this section or wider manuscript. I would consider removing it.
Ln754-768: Again, this text is interesting but is not relevant to this section. Consider removing it.
Ln782: “Exocytosis” is the wrong word. Do you mean “efflux”?
Ln785: This line is unintelligible; I have no idea what you are trying to say. Please rewrite for clarity.

Experimental design

No comment

Validity of the findings

Not applicable

---

## Round 0.3 · Minor Revisions

Boht Reviewers are happy with your responses to their comments. There are only a few minor editing issues to be corrected.

Reviewer 2 ·

Basic reporting

In the revision the authors took into consideration all the comments of the reviewers and the manuscript got well better to the original exceptionally good material. However, there are still some minor points that should be corrected.

Lane 276: Nietepski
Lane 331: conjugative
Lane 574: filter mating
Lane 733: correct the efflux pumps to increased expression of the cfiA gene

Experimental design

OK

Validity of the findings

OK

Additional comments

OK

Reviewer 3 ·

Basic reporting

I am broadly happy that the authors have addressed my concerns. Upon re-reading the manuscript I have identified a couple of other small errors that should be addressed. Please see my comments below:

Ln349: MTCE should be removed. MATE covers this family of efflux pumps.
Ln355: Exocytosis should be "efflux".
Ln531: Manipulator should be "operon"

Figure 1: The Y-axis should be labelled, I believe it should be millions.

Experimental design

NA

Validity of the findings

NA

Additional comments

NA

---

## Round 0.4 · accepted · Accept

The Authors have now fully addressed all the minor comments requested by the Reviewer.